# Fatigue Increases Muscle Activations but Does Not Change Maximal Joint Angles during the Bar Dip

**DOI:** 10.3390/ijerph192114390

**Published:** 2022-11-03

**Authors:** Alec McKenzie, Zachary Crowley-McHattan, Rudi Meir, John Whitting, Wynand Volschenk

**Affiliations:** Department of Sport and Exercise Science, Faculty of Health, Southern Cross University, Lismore, NSW 2480, Australia

**Keywords:** the dip, fatigue, exercise technique, exercise prescription, electromyography, kinematics

## Abstract

The purpose of this study was to profile and compare the bar dip’s kinematics and muscle activation patterns in non-fatigued and fatigued conditions. Fifteen healthy males completed one set of bar dips to exhaustion. Upper limb and trunk kinematics, using 3D motion capture, and muscle activation intensities of nine muscles, using surface electromyography, were recorded. The average kinematics and muscle activations of repetitions 2–4 were considered the non-fatigued condition, and the average of the final three repetitions was considered the fatigued condition. Paired t-tests were used to compare kinematics and muscle activation between conditions. Fatigue caused a significant increase in repetition duration (*p* < 0.001) and shifted the bottom position to a significantly earlier percentage of the repetition (*p* < 0.001). There were no significant changes in the peak joint angles measured. However, there were significant changes in body position at the top of the movement. Fatigue also caused an increase in peak activation amplitude in two agonist muscles (pectoralis major [*p* < 0.001], triceps brachii [*p* < 0.001]), and three stabilizer muscles. For practitioners prescribing the bar dip, fatigue did not cause drastic alterations in movement technique and appears to target pectoralis major and triceps brachii effectively.

## 1. Introduction

The dip is a popular bodyweight exercise frequently used to strengthen the muscles of the upper limb and trunk, and more specifically, the triceps brachii (TB) and pectoralis major (PM) [1,2]. There are many technique variations of the dip which can either decrease movement complexity (i.e., the bench dip) or increase movement complexity (i.e., the ring dip), which have recently been shown to exhibit differing neuromechanical profiles [3]. However, the most common variation appears to be the bar dip [4].

The bar dip, as depicted in Figure 1, has previously been prescribed to increase upper body push strength and muscular endurance [2,4,5]. More generally, dip variations are prescribed for “prehabilitation” and rehabilitation of upper body injury [6,7,8,9,10] and to increase upper extremity strength and power [11]. Despite its widespread popularity, there is little evidence justifying the use of the bar dip in such rehabilitation or performance programs.

When the bar dip is prescribed in the above contexts, it is common for a high volume of repetitions to be prescribed. For example, the maximal number of repetitions [5] and sets of 10–40+ repetitions [1,2] have been suggested in the literature for the purpose of increasing muscular endurance. These repetition schemes are likely to induce some level of fatigue, with a concomitant exercise-induced reduction in force or power [12]. However, no research has been conducted investigating the effects of fatigue on normal kinematics or muscle activation patterns when performing the bar dip.

The effects of fatigue have previously been investigated in other upper body push exercises such as the bench press [13,14,15] and push-up [16,17] and have been shown to significantly alter repetition characteristics such as increasing the duration of the upwards phase of these movements [13,14,18,19] and decreasing movement control [14]. These alterations in movement kinematics are often accompanied by changes in muscle activation strategies and coordination dynamics [12,20], with fatigue being characterized by an increase in amplitude and a decrease in spectral frequencies in an electromyography signal [21]. Some of these fatigue-induced changes may represent a practical concern for exercise professionals and exercisers when applied to the dip. Dips are completed some height above the ground and decreased movement control may increase fall risk, forcing the shoulder beyond the maximal range of motion, resulting in traumatic injury. Any potential increase in the risk of injury warrants investigation, particularly when considering that there are currently un-investigated practitioner concerns for a suspected high risk of injury to the shoulder when completing dip repetitions [22].

One more important potential fatigue-related control issue is the end-range shoulder extension observable at the bottom position when completing the dip. It has been suggested that there is an increased risk of injury to the anterior shoulder capsule and PM when in this position under load [22,23,24,25], especially when fatigued. This line of thinking is related to the perceived decreased neuromuscular and coordination control when fatigued which may put performers of the dip in an injurious position. However, these claims of fatigue-induced coordination dynamics change, and their relation to potential injury risk have minimal supporting evidence, with much of these reports coming from case reports [23,25], anecdotal examples [26,27], and unsupported claims [24]. As a result, having a better understanding of the neuromechanical profile of the dip will provide practitioners with evidence to more appropriately prescribe this exercise within different athletic contexts and populations.

The purpose of this study was to profile and compare the kinematics and muscle activation patterns of the bar dip in two conditions: (i) non-fatigued and (ii) fatigued. It was hypothesized that fatigue would cause an increase in the muscle’s electrical activity and that there would be significant changes in the peak joint angle’s experiences, particularly in the bottom position of the movement. An understanding of the neuromechanical profile of the dip will firstly improve the justification of its use in certain contexts and provide some rationale for the perceived injury risks associated with the exercise.

## 2. Materials and Methods

### 2.1. Participants

An a priori calculation found that a minimum sample size of 12 participants was needed to achieve a power of 0.8, with an alpha level of 0.05. Fifteen healthy males volunteered for this study (height = 172.05 ± 27.17 cm, weight = 87.19 ± 27.20 Kg, age = 29.07 ± 6.57 years, and 3.79 ± 5.23 years of regular resistance training experience). All participants regularly incorporated bodyweight dips, or variations thereof, in their weekly structured strength and conditioning programs. Participants were excluded if they had any current or major injuries to the upper limb or trunk in the 12 months prior to participation. This study was approved by the institution’s human research ethics committee (ECN-19-223). Participants completed an average of 24.93 ± 7.01 repetitions, which is comparable to collegiate lacrosse players who completed 25.17 ± 7.02 repetitions [5].

### 2.2. Procedures

All dips were performed on a V-shaped dip bar (Iron Edge, St Kilda, VIC, Australia) that was mounted on a free-standing matrix rack (Iron Edge, St Kilda, VIC, Australia) and anchored to the laboratory floor. The laboratory temperature was set to 24 °C for all participants. The matrix rack was surrounded by 14-3D motion capturing cameras (Vicon, Oxford, UK) which sampled data (200 Hz.) using a modified marker set of the University of Western Australia’s full-body model [28,29].

Nine surface electromyography (sEMG) electrodes (Delsys Avanti Wireless, Natick, MA, USA) were placed on the muscles of the participant’s right upper limb and trunk. These were dry, passive electrodes with a fixed spacing of 10 mm. The muscles were selected as follows:(i)Potential agonists of shoulder flexion and elbow extension: PM clavicular head, anterior deltoid (AD), and TB long head,(ii)Potential scapula stabilizers: upper trapezius (UT), serratus anterior (SA), and lower trapezius (LT), and(iii)Potential glenohumeral stabilizers: biceps brachii (BB), infraspinatus (IS), and latissimus dorsi (LD).

The skin was first prepared by shaving, abrading, and cleaning with an alcohol wipe before the electrodes were placed on standardized sites (Criswell, 2011). The sEMG data were sampled at 2000 Hz and synchronized with the kinematic data using Nexus data collection software version 2.4 (Nexus 2, Oxford, UK).

### 2.3. Data Collection

After signing an informed consent document, participants were familiarized with the equipment, laboratory, and testing protocols. Next, the retroreflective markers and sEMG electrodes were secured to the participant.

Following a standardized warm-up consisting of fifteen arm-swings, five rows, and ten push-ups [30], participants completed a maximal range of motion (ROM) test. This test was conducted with the dip bar at waist height and the participant’s feet planted on the ground to support their weight. The participant was instructed to lower themselves as far as possible, as if they were completing a dip, and to hold the bottom position for three seconds before returning to a standing position. Three repetitions were performed with the peak shoulder extension angle achieved, as measured through the 3D motion capture system, used as an indicator of maximal passive shoulder extension ROM.

Finally, participants moved the dip bar to a self-selected height (typically the waist to shoulder height) and were asked to complete one set of bar dips until volitional exhaustion. The research staff provided verbal encouragement throughout. Research staff offered no coaching cues or technique instruction at any point throughout a participant’s involvement, ensuring the technique and ROM used were entirely self-moderated.

### 2.4. Data Analysis

The vertical displacement of the mid-pelvis, which was determined geometrically from adjacent pelvis markers, was measured to determine dip depth and height, and to differentiate repetitions. The frame prior to the mid-pelvis lowering from its initial peak was deemed the starting position of each repetition. The participant then lowered their body through the downward phase to the bottom position, which was determined as the time point coinciding with the lowest mid-pelvis position. They then raised the body, through the upward phase, to the next peak in mid-pelvis height which was used to define the end position of one repetition. The final repetition was not counted if the participant did not fully extend their elbows at the end of the upward phase.

Raw kinematic data were processed using a fourth-order low-pass Butterworth filter (*f*_c_ = 6 Hz) before being time-normalized, represented in 0.5% increments of a total repetition. Next, the average at each 0.5% of the dip was calculated across the three repetitions in each dip condition (non-fatigued and fatigued). The kinematic variables of interest included the peak joint angle, and the angular joint position at the starting position, bottom position, and end positions for: extension, abduction, and external rotation of the shoulder, anterior thoracic lean (relative to vertical), and elbow flexion. The timing of when the peak joint angle occurred, represented as both a percentage of total repetition and as a percentage of repetition away from the bottom position, was also analyzed. All kinematic variables were compared between the two conditions.

The kinematic data were also used to determine the repetition characteristics such as: repetition duration, the rest period between repetitions, the relative timing of when the bottom position occurred (percentage of repetition), and the vertical range of the dip as measured by the overall vertical displacement of the mid-pelvis. All repetition characteristics were compared between conditions.

Raw sEMG data were smoothed and rectified using a root mean squared (RMS) algorithm, with a 0.4 s moving average. The RMS data were then normalized to the percentage of dip repetition and averaged across the three repetitions in each condition to match the processed kinematic data. For all muscles tested, the peak activation amplitude and the relative activation intensity at the point of maximal shoulder extension (normalized to a percentage of the peak amplitude that occurred during the non-fatigued condition, i.e., 100%) were determined and compared between conditions.

### 2.5. Statistical Analysis

All data were checked for normality using the Shapiro-Wilk test and the majority were found to be normally distributed. For normally distributed data, a paired *t*-test was used to investigate the differences in kinematic and muscle activation variables between the non-fatigued and fatigued conditions. For not-normally distributed data, a Wilcoxon signed-rank test (non-parametric t-test was used) and indicated in the presented results. Significance was set at an alpha level of 0.05, with Cohen’s *d* being calculated to measure the effect size between conditions (*d* > 0.2 small effect, *d* > 0.5 medium effect, *d* > 0.8 large effect, *d* > 1.2 very large effect [31]). All statistics were calculated using SPSS version 25 (IBM, Armonk, NY, USA).

## 3. Results

### 3.1. Repetition Characteristics

The repetition duration was significantly longer in the fatigued condition (non-fatigued = 1.35 ± 0.32 s, fatigue = 2.24 ± 0.66 s, *p* < 0.001, *d* = 1.72). There was also a significant increase in the rest time between repetitions in the fatigued condition (non-fatigued = 0.01 ± 0.03 s, fatigued = 0.45 ± 0.48 s, *p* = 0.002, *d* = 1.29). The bottom position occurred significantly earlier once fatigued, occurring at 52.96 ±4.01% of the dip repetition non-fatigued, and 44.19 ± 6.56% once fatigued (*p* < 0.001, *d* = 1.61). The vertical displacement of the mid-pelvis was significantly reduced once fatigued (non-fatigued = 351.68 ± 67.71 mm, fatigued = 311.00 ± 57.44 mm, *p* = 0.008, *d* = 0.65). Four participants attempted an additional repetition and were unsuccessful, these repetitions were not used in the analysis of the fatigued condition.

### 3.2. Kinematics

Peak joint angles and the angular position of each joint in the starting position, bottom position, and end position are presented in Table 1. There were no significant differences in any peak joint angles between the two conditions (*p* = 0.314–0.917, *d* = 0.01–0.26). In the starting position, there was greater shoulder extension (*p* = 0.003, *d* = 0.50) and external rotation (*p* = 0.043, *d* = 0.46) in the fatigued condition, whereas all other angles were similar (*p* = 0.059–0.270, *d* = 0.13–0.48). There were no significant differences in the joint angles at the point of maximal shoulder extension (*p* = 0.074–0.904, *d* = 0.03–0.43). In the end position, there were significant differences in increases in extension (*p* = 0.004, *d* = 0.55), abduction (*p* = 0.032, *d* = 0.51), and external rotation (*p* = 0.007, *d* = 0.66) of the shoulder once fatigued, but no differences in anterior thoracic lean (*p* = 0.201, *d* = 0.47) or elbow flexion (*p* = 0.982, *d* < 0.01).

During the maximal ROM trial, participants had a maximal shoulder extension angle of 87.3 ± 9.36°. When displayed as a percentage of the maximal ROM trial, participants used 75.96 ± 10.71% and 76.30 ± 11.13% of their observed maximal passive shoulder extension ROM in the non-fatigued and fatigued conditions respectively, these were not significantly different (*p* = 0.905, *d* = 0.03).

The peak joint angles occurred at a significantly different portion of the dip repetition for all joints (*p* < 0.001–0.004, *d* = 1.26–1.78) excluding abduction (*p* = 0.105, *d* = 0.71). However, when peak timing was normalised to the percentage of dip repetition relative to the bottom position, there was only one significant difference. This was peak elbow flexion, which occurred at 0.10 ± 0.67% after the bottom position in the non-fatigued condition and 1.29 ± 1.85% after the bottom position in the fatigued condition (*p* = 0.019, *d* = 0.86). Anterior thoracic lean, shoulder extension, and external rotation occurred at a maximum of ±2.47% away from the bottom position in both conditions and was not significantly different (*p* = 0.236–0.308, *d* = 0.34–0.48), whereas peak abduction occurred 10.97 ± 13.95% before the bottom position in the non-fatigued condition and 7.74 ± 20.70% before and during in the fatigued condition and was not significantly different (*p* = 0.666, *d* = 0.18).

Of the four failed repetitions not included in the above analysis, two participants achieved less peak shoulder extension in the final failed repetition compared to the fatigued condition (participant A: fatigued = 64.53°, failed = 60.9°, and participant B: fatigued = 84.78°, failed = 80.6°). The other two participants far exceeded their fatigued peak shoulder extension in the failed repetition, with both participants also exceeding their maximal ROM determined by the ROM test (participant C: fatigued = 68.85°, failed = 82.5°, maximal ROM test = 74.8°, and participant D: fatigued = 71.0°, failed = 80.0°, maximal ROM test = 77.8°).

### 3.3. Muscle Electromyography (Activations)

Figure 2 illustrates the peak sEMG amplitudes for all muscles tested. Five muscles (PM, TB, SA, LT, and LD) increased their peak amplitude during the fatigue condition (*p* < 0.001–0.043, *d* = 0.28–1.56). Four muscles (AD, UT, IS, and BB) had similar peak amplitudes in both conditions (*p* = 0.088–0.699, *d* = 0.05–0.39). At the point of maximal shoulder extension, three muscles (PM, SA, and LT) significantly increased their relative activation intensity during the fatigued condition, whereas BB exhibited significantly decrease muscle activity during fatigue (see Table 2).

## 4. Discussion

Overall, the kinematic characteristics of the dip were not different between fatigued and non-fatigued conditions. Fatigue did, however, cause significant increases in some of the muscles tested as hypothesized. Furthermore, there were significant changes in the repetition characteristics including an increase in repetition duration, an increase in rest time between repetitions, and a shift in the bottom position to an earlier point within the total repetition. Fatigue caused an increase in dip repetition duration, associated with a prolonged upward phase (when agonist muscles are acting concentrically) and the shift in the bottom position to an earlier portion of the repetition. Similar findings have been observed in many resistance exercises, such as the bench press [15,32], deadlift [33], and the squat [34], and can likely be explained by the phenomena of the ‘sticking’ point. The sticking point can be broadly defined as the point, or region, of an exercise where there is a disproportionately large reduction in movement velocity and an increase in movement difficulty [35]. While many factors may influence the sticking point, such as a reduced mechanical advantage [36,37], the most significant influence present in the current study is likely the onset of fatigue. As the agonist muscles fatigue, there is a reduction in the ability of these muscles to control the movement throughout both the downward (eccentric) and upward (concentric) phases; this is seen with the earlier arrival of the bottom position, the prolonging of the upward phase, and thereby the time spent within the sticking point. This could be of practical importance as there may be an increased likelihood of technique failure and subsequent injury within the region of the sticking point [36,38]. However, as participants were able to maintain consistent peak joint angles in both conditions this slight reduction in movement control and repetition characteristics does not appear to be detrimental when one set of bar dips is prescribed to volitional exhaustion. Despite these changes in repetition characteristics, a key finding of this investigation was that there were no significant differences in any peak joint angles nor the joint angles at the point of maximal shoulder extension for any joint motion investigated. This is interesting because fatigue has previously been shown to reduce shoulder [20,39,40] and elbow [41] joint position sense which is likely to be a controlling factor of dip depth. Following the fatigue of a single muscle group, healthy individuals are able to substantially alter the kinematic strategy of multi-jointed actions in order to maintain end-point accuracy [40]. In the current study, seemingly marginal kinematic changes occurred following the fatigue of multiple muscle groups. This indicates that participants were able to compensate for fatigue-induced deficits at multiple joints while maintaining a consistent technical standard. Due to this, it appears the increased concern of injury due to fatigue may not be warranted. While participants compensated at the bottom of the movement, some changes were observed at the top positions, although it should be noted that these positions do not place joints, with typically vulnerable structures, at compromised lengths for increased injury risk..

The vertical displacement of the mid-pelvis was significantly reduced once fatigued. As the peak joint angles were similar, this reduction in vertical displacement has largely resulted from a reduction in the peak height of the dip, as opposed to dip depth (i.e., the bottom position). This can be somewhat explained by an increase in shoulder extension in the starting and ending positions during the fatigued condition. As the technique used was self-moderated, it appears that participants may have subconsciously used elbow angle as an identifier of repetition completion regardless of the shoulder angle. However, while the shoulder angle at these two positions was significantly different between the two conditions, the average change was only 4.33° which may not entirely explain the overall reduction in mid-pelvis displacement. Rather, other variables not included in this study’s analysis may play contributing roles. For example, fatigue during the bench press has previously resulted in a reduction in bar height, despite no change in elbow angle at the top of the movement [14]. These researchers postulated that the height reduction was caused by a decrease’s scapula protraction due to fatigue of the scapula protractors, i.e., SA. The scapula kinematics throughout a dip repetition warrants future investigation due to the association of altered scapula kinematics and injury risk [42,43]. Although not assessed, a reduction in scapula depression (an increase in scapula elevation) may have been present in the current study, supported by fatigue being indicated (via an increase peak activation amplitude) in all muscles investigated with possible roles in scapula depression, i.e., LT, LD, SA, and PM.

Fatigue can be indicated by an increase in sEMG amplitude [12,21], likely due to an increase in motor unit recruitment. With regard to the peak activation amplitudes of the suspected agonists, some level of fatigue was present in the TB and PM but not the AD. The dip is primarily prescribed to target TB and PM [1,2] rather than the AD; however, due to the AD’s function in shoulder flexion [44] it is a likely agonist for the dip. The significant increases in TB and PM activation support the use of the bar dip in strength and conditioning programs when specifically targeting these muscles. In addition, the heightened activation of stabilizer muscles (SA, LT, and LD) indicate that the dip may be an appropriate in rehabilitation context when increasing strength of such muscles is required. However, further investigation is required within clinical populations.

A final finding of this investigation was that four participants failed their final repetition. Two of these participants exceeded their maximal ROM as conducted by the maximal shoulder extension ROM test. It is important to recognize that this is an incidental finding and the generalizations of athletic populations should be done with caution. However, it is also important to acknowledge the risk of traumatic events occurring, such as falls, when completing supramaximal dip repetitions. Exercise professionals should consider the use of a spotter and lowering the height of the dip bar, in addition to considering the frequency that the bar dip is prescribed to volitional exhaustion.

It is important to note the limitations in the practical applications of this investigation’s findings, the first being that all participants completed the dip under self-moderated conditions, meaning that if any strict coaching cues or technique standards are enforced, the neuromechanical profile of the dip may be altered. The second limitation is that the data was only analyzed from specific key time-points within the overall dip repetition, rather than assessing the repetition in its entirety as other data analysis methods accomplish (e.g., statistical parametric mapping). However, as this is the first systematic investigation into the bar dip, observing how experienced exercisers perform the dip at specific moments throughout will significantly improve the understanding and the prescription of this exercise. Finally, caution should be used when inferring the results of the current study to beginner exercisers. The participants in this study were experienced and regularly incorporated the dip into their training and therefore may exhibit differing neuromechanical profiles to beginners. Due to the large ranges of motion used during the dip, it is possible that the sEMG electrodes recognized signals from neighboring muscles; this is of particular concern regarding IS.

## 5. Conclusions

The bar dip appears to be an effective exercise for activating TB and PM. When prescribing the bar dip in practice, repetition schemes that induce fatigue may be beneficial in increasing the peak sEMG amplitudes of agonist muscles (TB and PM) and some stabilizer muscles (SA, LT, and LD) whilst having minimal effect on the overall movement’s kinematics. Indicating that bar dips to fatigue may be an effective training tool in strengthening the TB and PM, or the muscles of scapula depression. Due to the lack of kinematic differences, increased concerns for the dip when conducted to fatigue may not be warranted when prescribed in a single set. However, there is an inherent risk of falls during a dip, which should be considered when prescribing the dip in multiple sets to fatigue.

## Figures and Tables

**Figure 1 ijerph-19-14390-f001:**
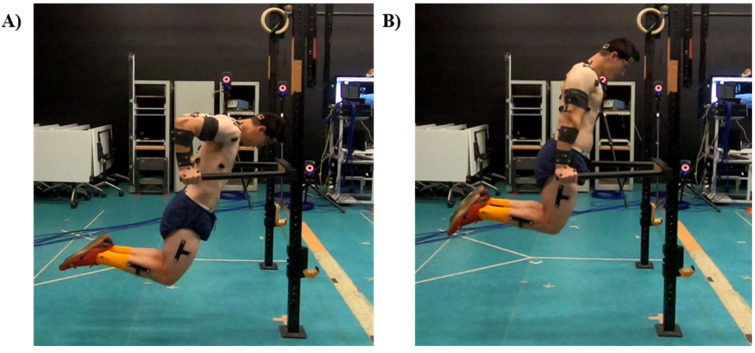
A typical bar dip with, (**A**) being the bottom position, and (**B**) being the start/ending position.

**Figure 2 ijerph-19-14390-f002:**
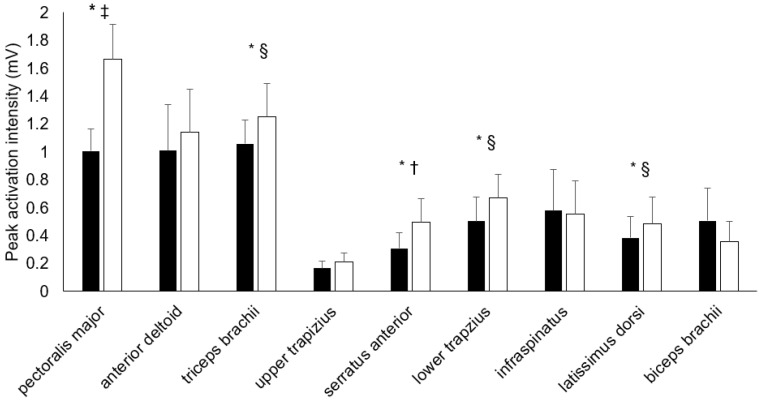
Mean peak sEMG amplitudes for each muscle during the bar dip in non-fatigued (black) and fatigued (white) conditions. * indicates a significantly greater muscle activation intensity in the fatigued condition, § indicates a small effect size (*d* > 0.2), † indicates medium effect size (*d* > 0.8), ‡ indicates a very large effect size (*d* > 1.2). A Wilcoxon signed-rank test was used for the upper trapezius, serratus anterior, and biceps brachii.

**Table 1 ijerph-19-14390-t001:** Mean (±SD) peak joint angles, and joint angles at the starting position, point of maximal shoulder extension, and end position during the bar dip in non-fatigued and fatigued conditions.

	*Peak Joint Angles (* ° *)*		*Joint Angle at the Starting Positions (* ° *)*		*Joint Angles at the Point of Maximal Shoulder Extension (* ° *)*		*Joint Angles at the End Position (* ° *)*	
Joint Movement	Non-Fatigued	Fatigue	Non-Fatigued	Fatigue	Non-Fatigued	Fatigue	Non-Fatigued	Fatigue
Shoulder extension	67.66 ± 11.34	67.18 ± 10.06	2.99 ± 8.21 *†	7.18 ± 8.67 *†	67.66 ± 11.34	67.18 ± 10.06	3.51 ± 7.97 *†	8.07 ± 8.59 *†
Abduction	31.25 ± 3.56	32.84 ± 9.32	12.07 ± 4.09	12.66 ±5.24	27.86 ±11.35	26.78 ±5.91	11.88 ± 4.02 *†	14.29 ± 5.32 *†
External rotation	9.03 ± 14.30	8.83 ± 13.90	−41.96 ± 14.29 *§	−46.61 ± 9.94 *§	8.98 ± 14.29	8.75 ± 13.92	−42.32 ± 10.70 *†	−49.93 ± 11.11 *†
Anterior thoracic lean	37.25 ± 9.78	35.93 ± 7.38	18.49 ± 12.54	13.24 ±9.13	30.34 ± 21.06	34.61 ± 5.89	17.74 ± 11.87	12.76 ± 8.92
Elbow flexion	114.63 ± 8.59	112.20 ± 10.25	18.20 ± 11.04	15.55 ± 12.57	115.88 ± 9.72	111.59 ± 10.05	17.37 ± 11.35	17.72 ± 14.13

Note: * indicates a significant difference (*p* < 0.05), § indicates small effect size (*d* > 0.2), † indicates medium effect size (*d* > 0.8). Peak external rotation and external rotation at the point of maximal shoulder extension were compared between conditions using the Wilcoxon singed-rank test as the data was not normally distributed, all other variables were normally distributed and were therefore compared using a paired *t*-test.

**Table 2 ijerph-19-14390-t002:** Maximal (mean ±SD) Activation intensities at the point of maximal shoulder extension.

Muscle	Non-Fatigued	Fatigued	*p*-Value (Cohen’s *d*)
*Activation intensity at the point of maximal shoulder extension* *(displayed as % of peak activation during the non-fatigued condition)*			
Pectoralis major	64.80 ± 9.55 *	105.68 ± 36.17 *	<0.001 (1.55)
Anterior deltoid ^a^	66.04 ± 22.96	59.35 ± 31.32	0.552 (0.24)
Triceps brachii	79.53 ± 12.38	78.37 ± 20.71	0.832 (0.07)
Upper trapezius	74.77 ± 12.38	89.93 ± 68.62	0.352 (0.30)
Serratus anterior	65.30 ± 22.78 *	145.17 ± 125.81 *	0.032 (0.88)
Lower trapezius	77.39 ± 17.16 *	101.52 ± 31.77 *	0.010 (0.95)
Infraspinatus	67.85 ± 20.79	78.38 ± 40.41	0.291 (0.33)
Latissimus dorsi	67.86 ± 24.33	72.59 ± 27.39	0.427 (0.18)
Biceps brachii ^a^	69.01 ± 32.07 *	61.01 ± 41.46 *	0.046 (0.22)

*Note*: * Indicates a significant difference (*p* < 0.05). ^a^ indicates where a Wilcoxon signed-rank test was used. A paired samples *t*-test was used for all other variables.

## Data Availability

All data that support these findings are available from the corresponding author upon reasonable request.

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
