# Peer review of "Fatigue Increases Muscle Activations but Does Not Change Maximal Joint Angles during the Bar Dip"

_ijerph, 2022, doi:10.3390/ijerph192114390_

Round 1
Reviewer 1 Report
It is an interesting manuscript, but it is not clear how long the fitness routines.
If the room temperature is a relevant factor or the subject's age, etc., could this description be improved?
It is said "... Cohen’s d being calculated to measure the effect size" but it is not said which d value was used
Author Response
- It is an interesting manuscript, but it is not clear how long the fitness routines
Participants completed one set of bar dips to volitional exhaustion. The time was not controlled, so as to allow for the capture of each participant’s natural repetition rate and therefore natural technique under each condition, Participants completed an average of 24.94 ±7.01 repetitions (line 86).
- If the room temperature is a relevant factor or the subjects age etc., could this description be improved?
The authors agree that room temperature may a relevant factor and therefore it was controlled. The room thermostat was set to 24°C for all participants – line 92 added. The authors believe that age may not be as relevant as training age and experience completing the bar dip. The age ranged from 20-40 years with a mean of 29.07 ±6.57 years. This information can be found on line 80 and is now listed in the limitations (comparing our participants to beginner exercisers) in lines 340-343.
- It is said “…Cohens d is being calculated to measure effect size” but it is not said which d value was used
The authors are not sure what the reviewer is suggesting with this comment. We are not aware of any other versions of Cohen’s d. The standard Chone’s d has been reported in typical format. All effects sizes have been reported in the results section.
Reviewer 2 Report
I appreciate some important facts, that are not so well-known to the clinicians. I have some comments:
In results, the findings are well ordered (repetitions, kinematics, sEMG), but discussion is less strict. So the reader has problems to follow the discussion.
Some facts are suprising and need more discussion and arguments (fatigue and prolonged duration of contraction, fatigue and more electrical activity.etc.)
Type of muscles (fibres I or II) and characterization of fatigue
Fatigue in trained sportsmen and comparing with beginners
Author Response
- In the results, the findings are well ordered (repetitions, kinematics, sEMG), but discussion is less strict. So the reader has problems to follow the discussion.
The authors have improved the statements linking paragraphs (including lines 291-292). However, the authors feel subsections within the discussion would be inappropriate as this would not allow for the synthesis of kinematic and sEMG data, as per our discussion, to provide a coherent overview of the findings.
- Some facts are surprising and need more discussion and arguments (fatigue and prolonged duration of contraction, fatigue and more electrical activity etc.)
The authors agree these are interesting results. The increased repetition duration is discussed in lines 253-262. Fatigue and increased muscle activity was first described by De Luca (1984) and a small explanation of this is added to lines 313-314.
- Type of muscles (fibres I or II) and characterisation of fatigue
The authors agree this is likely to have some impact on the characterisation of fatigue. However, this was outside of the scope of this study being the first investigation of the dip. It would certainly be an interesting aspect to be investigated in future research.
- Fatigue in trained sportsmen and comparing with beginners
The authors agree, and this has now been listed as a limitation of this investigation (lines 340-343).
Reviewer 3 Report
Dear Authors,
Thank you very much for the opportunity to review this paper.
The purpose of this study was to profile and compare the kinematics and muscle activation patterns of the bar dip in two conditions: i) non-fatigued and ii) fatigued. I have some doubts about the methodological correctness of the planned research.
Comments and Suggestions for Authors:
1. In my opinion, the size of the study group is very small to draw profound conclusions, even though the required minimum size for this type of analysis is even smaller.
2. So I have doubts about the fulfillment of the condition of the normality of the distribution of features. Please provide the numerical values of the test verifying the normality of the distribution of features. Please consider using nonparametric tests.
3. Due to the large variance in the basic somatic parameters (eg height and weight), would it not be advisable to perform analyzes in, for example, two subgroups (eg low body height versus high or heavy versus light). Additionally, in my opinion, body height and the highly correlated arm length (lever length) may have significant significance for the presented results (as well as body weight).
4. Thank you for the information on obtaining ethical consent for the conducted research.
5. Theoretical chapters well written.
6. Editorial errors in the preparation of the manuscript - to be corrected by the authors.
Thank you for the opportunity to review this article.
Author Response
Reviewer 3:
- In my opinion, the size of the study group is very small to draw profound conclusions, even though the required minimum size for this type of analysis is even smaller.
The authors agree that the sample size of 15 is a small limitation. However, our power analysis indicated that a minimum of 12 participants was needed to a achieve a power of 0.8 (lines 78-79), therefore allowing for a discussion of our findings in this context. As a first study of its kind with the dip movement, we believe these findings have sufficient statistical power to give readers confidence that they could be repeated.
- So I have doubts about the fulfillment of the condition of the normality of the distribution of features. Please provide the numerical values of the test verifying the normality of the distribution of features. Please consider using nonparametric tests
The authors acknowledge that not all data were normally distributed. However, of the more than 100 statistical tests performed, only 7 displayed non-normal distributions. Therefore, we ran parametric tests for simplicity and consistency. However, we recognize that it may have been more appropriate to run non-parametric tests on these 7 data sets, and have not used the Wilcoxon signed-rank test (non-parametric t-test) for those variables. These variables are included in the table below.
As displayed, this analysis has changed the result for one variable, being normalised biceps brachii activation at the point of maximal shoulder extension. Although these new statistics don’t alter our main finding or conclusion, we have adjusted the manuscript in the methods section (lines 162-169), and results section (lines 205-209, 252-253) accordingly.
|
Variable |
t-test p-value |
Wilcoxon p-value |
|
Upper trapezius maximum |
0.088 |
0.035 |
|
Serratus anterior maximum |
0.043 |
0.023 |
|
Biceps brachii maximum |
0.273 |
0.249 |
|
External rotation maximum |
0.857 |
0.492 |
|
External rotation at bottom |
0.856 |
0.492 |
|
Anterior deltoid at bottom |
0.455 |
0.552 |
|
Biceps brachii at bottom |
0.569 |
0.046 |
- Due to the large variance in the basic somatic parameters (e.g. height and weight(. Would it not be advisable to perform analyses in, for example, two subgroups (e.g. low body height versus high or heavy vs light). Additionally, in my opinion, body height and the highly correlated arm length (lever length) may have significance for the presented results (as well as body weight)
The authors agree that this is an interesting avenue for future research. Similar research has been conducted on the effects of limb length during a squat and has shown to be impactful. However, this is beyond the scope of the current investigation, being the first study to investigate the fatigue with dip neuromechanics. Further, sub-analysis using the current sample size would decrease the statistical power, likely preventing meaningful inferences to be made.
Round 2
Reviewer 3 Report
Dear Authors,
Thank you for the opportunity to review this paper.
Thank you for taking into account my comments and relevant significant additions.
Thank you for your response to my doubt. I understand your opinion and your point of view, and I accept these arguments. I hope that, the manuscript will receive great interest and citation).
Thank you.